# SARS-CoV-2 Omicron Variant, Lineage BA.1, Is Associated with Lower Viral Load in Nasopharyngeal Samples Compared to Delta Variant

**DOI:** 10.3390/v14050919

**Published:** 2022-04-28

**Authors:** Célia Sentis, Geneviève Billaud, Antonin Bal, Emilie Frobert, Maude Bouscambert, Gregory Destras, Laurence Josset, Bruno Lina, Florence Morfin, Alexandre Gaymard

**Affiliations:** 1Laboratoire de Virologie, Institut des Agents Infectieux, Laboratoire Associé au Centre National de Référence des Virus des Infections Respiratoires, Hospices Civils de Lyon, F-69004 Lyon, France; celia.sentis01@chu-lyon.fr (C.S.); genevieve.billaud@chu-lyon.fr (G.B.); antonin.bal@chu-lyon.fr (A.B.); emilie.frobert@chu-lyon.fr (E.F.); maude.bouscambert-duchamp@chu-lyon.fr (M.B.); gregory.destras@chu-lyon.fr (G.D.); laurence.josset@chu-lyon.fr (L.J.); bruno.lina@chu-lyon.fr (B.L.); florence.morfin-sherpa@chu-lyon.fr (F.M.); 2CIRI, Centre International de Recherche en Infectiologie, Team VirPath, Univ Lyon, Inserm, U1111, Université Claude Bernard Lyon 1, CNRS, UMR5308, ENS de Lyon, F-69007 Lyon, France; 3GenEPII Platform, Institut des Agents Infectieux, Hospices Civils de Lyon, F-69004 Lyon, France

**Keywords:** SARS-CoV-2, COVID-19, viral load, Delta variant, Omicron variant

## Abstract

Objectives: High viral load in upper respiratory tract specimens observed for Delta cases might contribute to its increased infectivity compared to the other variant. However, it is not yet documented if the Omicron variant’s enhanced infectivity is also related to a higher viral load. Our aim was to determine if the Omicron variant’s spread is also related to higher viral loads compared to the Delta variant. Methods: Nasopharyngeal swabs, 129 (Omicron) and 85 (Delta), from Health Care Workers were collected during December 2021 at the University Hospital of Lyon, France. Cycle threshold (Ct) for the RdRp target of cobas^®^ 6800 SARS-CoV-2 assay was used as a proxy to evaluate SARS-CoV-2 viral load. Variant identification was performed using a screening panel and confirmed by whole genome sequencing. Results: Herein, we showed that the RT-PCR Ct values in Health Care Workers sampled within 5 days after symptom onset were significantly higher for Omicron cases than Delta cases (21.7 for Delta variant and 23.8 for Omicron variant, *p* = 0.008). This difference was also observed regarding patient with complete vaccination. Conclusions: This result supports the studies showing that the increased transmissibility of Omicron is related to other mechanisms than higher virus excretion.

## 1. Introduction

At the end of 2020, the first SARS-CoV-2 variant of concern (VOC), named Alpha, was detected and became the main variant just a few months after [1,2]. Then, Delta variant firstly detected in March 2021 rapidly spread worldwide and became the major variant during the second part of 2021. These VOCs demonstrated increased infectivity that was related to better affinity for Angiotensin-Converting Enzyme 2 (ACE2) cellular receptor and higher viral load in respiratory tract samples [3,4,5,6,7].

During the last trimester of 2021, a new variant named Omicron emerged and was immediately classified as a VOC due to the large number of mutations found in the spike protein, including several mutations known to be associated with higher transmissibility and/or immune escape [8,9]. The Omicron variant is the most contagious form of SARS-CoV-2 known so far and became dominant worldwide in a few weeks [10]. However, it is not yet documented if its enhanced infectivity is also related to a higher viral load as reported for other variants [5,6].

## 2. Materials and Methods

To determine if the Omicron variant’s spread is related to higher viral loads compared to Delta variant, we retrospectively collected nasopharyngeal swabs from screening center dedicated to health care workers (HCW) and family at the University Hospital of Lyon, France. Data from samples taken between 1 December 2021 and 31 December 2021 were collected, a period with circulation of Delta variant and emergence of Omicron variants in Lyon, France (Appendix A). All samples with variant identification were included without any prior selection. Presence of symptoms, time since symptom onset, age, sex and vaccination status were collected for each patient. The vaccination status was divided into five categories according to French vaccination strategy: not vaccinated, partially vaccinated (1 dose or 1 infection), completely vaccinated (2 doses or 1 dose and 1 infection or less than 7 days after the third dose), boosted (3 doses or 2 doses and 1 infection) or unknown (data not available) (Table 1). 

The French national strategy includes a screening test specific for SARS-CoV-2 (antigenic and molecular assays), then positive samples are tested by RT-PCR targeting some mutations (E484K, L452R and K417N) and by whole genome sequencing. In our laboratory, SARS-CoV-2 screening were conducted only by molecular assays using the cobas^®^ 6800 SARS-CoV-2 assay (Roche, Bâle, Switzerland). Without calibration range for cobas^®^ 6800 SARS-CoV-2 assay, cycle threshold (Ct) for the RNA-dependant RNA polymerase (*RdRp*) target was used as a proxy to evaluate SARS-CoV-2 viral load and Ct differences was transformed using the theoretical equation: $\Delta CV=2^{\bar{Ct_{Group1}}-\bar{Ct_{Group2}}}$. Reproducibility of the cobas^®^ 6800 SARS-CoV-2 assay was controlled using homemade quality control across 33 different runs. Mutation screening was performed using TaqMan SARS-CoV-2 Mutation Panel (Thermofisher, Waltham, MA, USA) and whole genome sequencing with COVIDSeq assay (Illumina, San Diego, CA, USA). All statistical analyses were conducted using GraphPad Prism^®^ (version 8.0.2). Mean Ct difference between groups was assessed using the Student’s *t* test or Mann-Whitney test, as appropriate. Categorical variables were compared using the Chi-square test or Fisher’s exact test. 

## 3. Results

### 3.1. Cohort Description

Between 1 December 2021 and 31 December 2021, 214 patients were included after their Delta or Omicron infection diagnostic. These patients were mainly HCW under 50 years old (median age: 32 years old) with symptom (71%, n = 152/214) for less than 5 days (82%, n = 125/152). No symptom information was available, except for time since symptoms onset, but patients were included at a screening center so no severe cases were included. No differences were observed between patients infected with Omicron or Delta variant for sex, age and time since symptoms onset. However, vaccination status were different between our two groups. Delta infected patients had mainly a complete vaccination status (61.2%, n = 52/85) and Omicron infected patients had mainly boosted vaccination (45%, n = 58/129) (Table 1). 

### 3.2. Viral Load Analyses

Using all the cohort, patients infected with Omicron variant had a lower viral load (higher Ct value) compared to patients infected with Delta variant (22.6 for Delta vs. 24.4 for Omicron, *p* = 0.004) (Figure 1A). Omicron-infected patients with symptom onset less than 5 days before sampling had a lower viral load compared to patients infected with Delta variant (23.8 and 21.7, respectively, *p* = 0.008). Moreover, regardless of the vaccination status, the largest difference was found in patients with symptom onset under 1 day, with viral load almost 1 log_10_ lower with Omicron variant compared to Delta variant (21.3 for Delta vs. 24.2 for Omicron, *p* = 0.035) (Figure 1A and Appendix A). No differences were observed for asymptomatic patients and patients with symptoms onset over 5 days. Indeed, asymptomatic group is a heterogeneous group composed of patients that are either presymptomatic or at any time during infection. 

Regardless of the vaccination status, significant viral load differences were found only for patients over 40 years old (20.9 for Delta vs. 23.6 for Omicron, *p* = 0.006) (Appendix A, Figure 1B) with higher viral load found in patients infected by Delta variant. 

Finally, regarding patients that were completely vaccinated, the Omicron variant was characterized by a lower viral load compared to patients infected by the Delta variant (21.8 for Delta vs. 24.8 for Omicron, *p* = 0.001). In boosted patients the inverse trend was observed with a higher viral load during Omicron variant infection (25.8 for Delta vs. 23.8 for Omicron *p* = 0.08) (Figure 1C). No viral load differences were observed between completely vaccinated and boosted patients with Omicron infection (23.9 for complete vaccination and 23.3 for boosted, *p* = 0.009), but a lower viral load was found in boosted patients infected by Delta (20.7 for complete vaccination and 25.6 for boosted, *p* = 0.009). Of note, the proportion of third dose (boosted) was higher at the end of December compared to the beginning of December (8.0% week 48 vs. 54.6% week 52, Appendix A).

## 4. Discussion

For the previous Alpha and Delta VOCs higher transmission rates have been related to higher viral loads [5,6,7]. In contrast, our results showed a higher Ct value (+2.8), reflecting lower viral load (−0.85 log_10_), for patients infected by Omicron, compared to patients infected by Delta variant. This is in agreement with a recent study reporting patients follow-up after Delta and Omicron infection showing a peak viral load at a Ct value of 23.3 for Omicron and 20.5 for Delta [11]. In another study, Puhach et al., reported a low correlation between RNA genome copies and infectious virus shedding evaluated by viral culture [12]. Regarding Omicron, they showed a trend to lower viral load compared to Delta (RNA genome copies and infectious virus titers) that was not significant probably due to low patients number in the Omicron group (n = 18) [12]. A recent study from Laitman et al. reported lower viral load for Omicron compared to Delta variant but only in symptomatic patients tested by the Abbott platform. However, they could not evaluate the time since symptoms onset nor the vaccination status that could affect viral loads [13]. These data combined with ours could suggest that higher infectiousness of Omicron may not be related to an increased viral load as reported for previous variants. Complete mechanisms driving the higher transmissibility of the Omicron variant are still unknown. Infectiousness could be multifactorial and related to background immunity, respiratory symptoms (cough, sneeze), duration of viral excretion, age and viral parameters such as new viral entry mechanism [11,14,15].

The present study has several limitations as viral load was estimated by Ct without normalization. Indeed, without a calibration range, Ct analysis is only a semi-quantitative evaluation of viral load. However, analysis of 33 positive controls showed a very good assay reproducibility (mean Ct: 33.8 ± 0.34). Thus, Ct differences observed between Delta- and Omicron-infected patients is significant and not related to the assay. Viral load differences were found only for symptomatic patient with time to symptoms onset under 5 days. This may be explained by a slower viral load decrease for patients infected with Omicron, but we were unable to confirm this observation due to low number of patients included over 5 days after symptoms onset (12%, n = 27/214). The viral load differences were not found in the asymptomatic group but this group represent very heterogeneous cases with patient sampled at any time during their infection. We previously recorded a wide variability of viral load in asymptomatic patients (variability increased by 15% compared to different groups of symptomatic patients, data not published). The viral load differences were also found significant only for patient over 40 years old but the same trend is observed for patient under 40 years old. This observation could be linked either to the percentage of asymptomatic patients under 40 (75.8 and 67.6% in Delta and Omicron group, respectively) or to the vaccine efficacy which is known to be higher in younger patients. 

Moreover, only the BA.1 lineage were circulating in France during the study period and results for other Omicron lineages such as BA.2 could differ. Our results were also impacted by the vaccination strategy. Omicron viral load was lower for patients with complete vaccination but the inverse trend was observed for boosted patients. This might be related to a lower susceptibility to neutralizing antibodies for the Omicron variant compared to the Delta variant. Boosted patients infected by the Delta variant might be more protected than patients infected by Omicron, which would explain the lower viral load in the Delta group [8,16,17]. These results should be taken carefully as more patients with boosted vaccination were observed in the Omicron infection group and counterwise more patients with complete vaccination were observed in the Delta infection group (Table 1). This observation has to be related to the National Vaccination Strategy, as the French government announced a mandatory third dose vaccination to health care workers during December. In addition, most of the positive samples for Delta variant were collected at the beginning of December. The Omicron variant began to be detected in France mid-December 2021 and overthrow the Delta variant during the last week of December (Appendix A). A more in-depth study taking into account vaccination status, time since the last injection and antibody levels will be necessary to better understand the impact of the immune response on the infection by different SARS-CoV-2 variants [11].

Finally, lower viral load during Omicron infection might impact viral diagnosis. Even if rapid antigenic testing is still able to detect Omicron, higher Ct values in Omicron cases may be associated with an increased number of false negative results compared to Delta. This must not preclude from using antigenic testing devices for screening SARS-CoV-2, but the interpretation may be cautious. Moreover, patient monitoring using Ct values should be cautiously interpreted according to each patient situation. Ct value could be a poor indicator of infectiousness, especially in presence of neutralizing antibodies. In the context of a largely vaccinated population, new criteria must be defined and new biomarkers have to be looked for.

## 5. Conclusions

Health Care Workers infected with Omicron variant (BA.1) have a lower viral load in nasopharyngeal samples compared to HCW infected with Delta variant (Ct = 23.8 and 21.7 respectively, *p* = 0.008). Thus, the Omicron variant’s increased transmissibility seems to be related to other mechanisms than a higher viral load. 

## Figures and Tables

**Figure 1 viruses-14-00919-f001:**
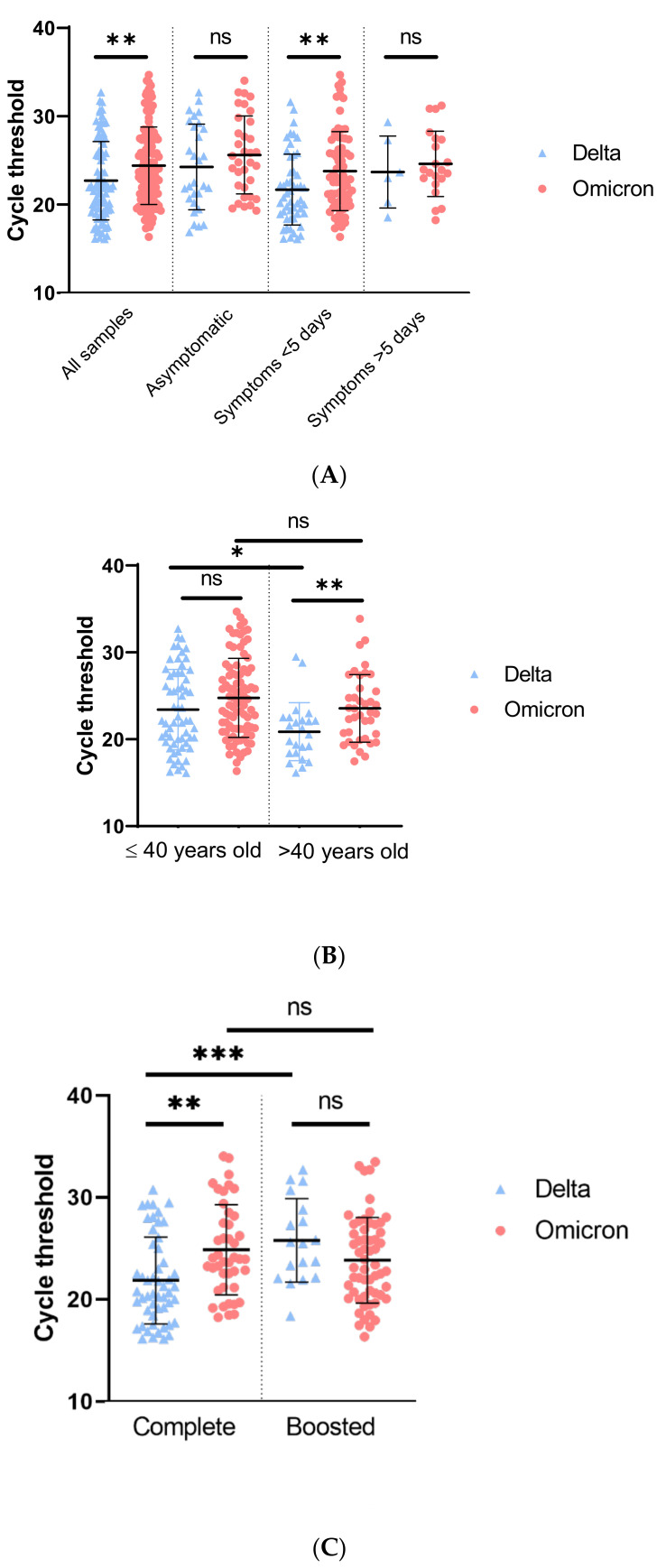
RT-PCR cycle threshold values for Delta or Omicron. (**A**): Cycle threshold of Delta or Omicron variant according to symptoms. Global analysis included all samples (Delta, n = 85, Omicron, n = 129). Asymptomatic patients (Delta, n = 28 and Omicron, n = 34), symptoms appearing less than 5 days prior sampling (Delta, n = 51 and Omicron, n = 74) and symptoms appearing more than 5 days prior sampling (Delta, n = 6 and Omicron, n = 21). (**B**): Cycle threshold of Omicron or Delta variant by age. Ct was analyzed for patients under 40 years old (Delta, n = 61 and Omicron, n = 90) and patients over 40 years old (Delta, n = 24 and Omicron, n = 39). (**C**): Cycle threshold of Delta or Omicron variant according to vaccination status. Vaccination was considered complete when patient received 2 doses or 1 dose and 1 infection or less than 7 days after the third dose (Delta, n = 52 and Omicron, n = 43) and boosted when patients received 3 doses or 2 doses and 1 infection (Delta, n = 18 and Omicron, n = 58). *p*-value was calculated with Student’s *t*-test. ns = not significant, * = <0.05, ** = <0.01, *** = <0.001.

**Table 1 viruses-14-00919-t001:** Demographic data.

	Delta (n = 85)	Omicron (n = 129)	*p*-Value
Sex			
Women	65.9% (56)	62.8% (81)	0.66
Men	34.1% (29)	37.2% (48)	
Index ratio	1.93	1.68	
Age			
20–30	37.7% (32)	45.8% (59)	0.26
31–40	34.1% (29)	24.0% (31)	0.12
41–50	16.5% (14)	18.6% (24)	0.72
>51 years old	11.7% (10)	11.6% (15)	1.0
Symptoms			
Asymptomatic	32.9% (28)	26.3% (34)	0.36
Day before or day of sampling	24.7% (21)	28.7% (37)	0.64
2 to 4 days before sampling	35.3% (30)	28.7% (37)	0.29
5 to 7 days before sampling	5.9% (5)	12.4% (16)	0.16
8 to 14 days before sampling	1.2% (1)	3.9% (5)	0.4
Vaccination status			
Not vaccinated	4.7% (4)	5.4% (7)	1.0
Partial vaccination	8.2% (7)	7.8% (10)	1.0
Complete vaccination	61.2% (52)	33.3% (43)	<0.0001
Boosted vaccination	21.2% (18)	45.0% (58)	0.0004
Unknown	4.7% (4)	8.5% (11)	0.41

## Data Availability

SARS-CoV-2 whole genomes sequenced in this study were deposited in the GISAID database.

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
