# Peer review of "SARS-CoV-2 Omicron Variant, Lineage BA.1, Is Associated with Lower Viral Load in Nasopharyngeal Samples Compared to Delta Variant"

_viruses, 2022, doi:10.3390/v14050919_

Round 1

Reviewer 1 Report

This is a compact, well performed and clearly written article that impresses with its clarity. The authors asked themselves one question and answered it very clearly and documented. From the mosaic of such specific questions and answers, a general picture of the properties of a new pathogen is formed. I have just a few comments.

Point 1: Line 77-79. The sentence “No difference were observed between patients infected with Omicron or Delta variant except for the vaccination status” is confusing. There is an opinion that Delta infection causes a more severe course of the disease than Omicron. This sentence sounds like a counterpoint to this opinion. Please, underline that this is just a description of the cohorts.

Point 2: Line 84-87. These two sentences have to be rephrased for clarity. For instance, “Omicron infected patients with symptoms appearing less than 5 days before sampling had a lower viral load compared to patients infected with Delta variant (Figure 1A).”

Point 3: Line 89. How virus titer in log10 was calculated? How did you evaluate infectious virus? Where are the data confirmed these calculations?

Point 4: Line 84-93. What is this all about? What particular groups were compared? For instance, does “asymptomatic group” include patients from all five categories according to French vaccination strategy? Such analysis is interesting and unusual; however, don't you think that you are defining the so-called “average temperature in the hospital”? I can understand if compare vaccinated subjects with nonvaccinated ones, or completely vaccinated with boosted persons. However, what information does a breakdown, for example, by age give, if these groups are mixture of vaccinated, nonvaccinated and boosted subjects? Of course, you can divide patients into groups as you see fit, just please explain why this is logical.

Point 5: The Discussion section is well written, but it is more than twice the size of the Results section. The results obtained by the authors are very interesting, and it makes sense to describe them in more detail.

Point 6: According to the Instructions for authors of VIRUSES, the Conclusions section is not mandatory, but it would be great to add it to the manuscript for clarity.

Point 7: Acknowledgments. This section is unfinished. Please, fix.

Author Response

Comments and Suggestions for Authors

This is a compact, well performed and clearly written article that impresses with its clarity. The authors asked themselves one question and answered it very clearly and documented. From the mosaic of such specific questions and answers, a general picture of the properties of a new pathogen is formed. I have just a few comments.

We thank the reviewers for these kind comments

Point 1: Line 77-79. The sentence “No difference were observed between patients infected with Omicron or Delta variant except for the vaccination status” is confusing. There is an opinion that Delta infection causes a more severe course of the disease than Omicron. This sentence sounds like a counterpoint to this opinion. Please, underline that this is just a description of the cohorts.

In agreement with reviewer 1, we modified the text between lines 79 to 87 as it is just a description of the cohort and not a clinical comparison.

We add information on the absence of clinical data and detailed our cohort description

Point 2: Line 84-87. These two sentences have to be rephrased for clarity. For instance, “Omicron infected patients with symptoms appearing less than 5 days before sampling had a lower viral load compared to patients infected with Delta variant (Figure 1A).”

In agreement with reviewer 1, we modified the sentence as proposed line 94 to 96.

Point 3: Line 89. How virus titer in log10 was calculated? How did you evaluate infectious virus? Where are the data confirmed these calculations?

Estimation of viral load in log10 is related to the relationship: 3.3 cycle threshold = 1 log10. During our work, cycle threshold was used as a proxy to evaluate viral load as there is no calibration range for cobas® 6800 SARS-CoV-2 assay. The exact difference is described line 137 and 138 of the manuscript, which is a difference of 2.8 cycle threshold which represent 0.85 log10 (log (22.8)). We choose to present this difference as “almost 1 log10” because without the calibration range this is an approximation. We wanted to be as clear as possible for reader whose are not familiar with semi-quantitative results from RT-PCR test.

We add some details on this calculation in the M&M section, line 65 to 69 :

“Without calibration range for cobas® 6800 SARS-CoV-2 assay, cycle threshold (Ct) for the RdRp target was used as a proxy to evaluate SARS-CoV-2 viral load and Ct differences was transformed using the theoretical equation: . Reproducibility of the cobas® 6800 SARS-CoV-2 assay was controlled using homemade quality control across 33 different runs.”

We did not evaluate infectious virus titer in our study.

Point 4: Line 84-93. What is this all about? What particular groups were compared? For instance, does “asymptomatic group” include patients from all five categories according to French vaccination strategy? Such analysis is interesting and unusual; however, don't you think that you are defining the so-called “average temperature in the hospital”? I can understand if compare vaccinated subjects with nonvaccinated ones, or completely vaccinated with boosted persons. However, what information does a breakdown, for example, by age give, if these groups are mixture of vaccinated, nonvaccinated and boosted subjects? Of course, you can divide patients into groups as you see fit, just please explain why this is logical.

All groups and subgroups are mixture from one antother. We try to separate the main variable that could influence viral load.

We selected to breakdown our patients using:

  • Age category because we add some internal data before the start of vaccination that tend to show that older patients have higher viral load. We also did a modelisation work (Néant N., et al. Modeling SARS-CoV-2 viral kinetics and association with mortality in hospitalized patients from the French COVID cohort. Proceedings of the National Academy of Sciences of the United States of America, 2021.) that showed slower decrease of viral load in patient over 65 years old. We discussed results found using this specific categorisation line 167 to 172.
  • Time since symptoms onset and asymptomatic patients. Asymptomatic patients represent very heterogeneous cases with patient sampled at any time during their infec-tion. Differences between VOC for transmissibility should be more important during the first day of the infection so we distinguish the acute phase (under 5 days) to the late phase of the infection (over 5 days). Discussed line 159 to 165.
  • Vaccination status is the main variable, which could affect the results comparing viral load. Line 175 to 190. 

We also used multivariate analyses and did some ACPmix to explore our data but we did not present these results as more subgrouping lead to very few patients per group and we could only see trend, which were concordant with our main results.

Point 5: The Discussion section is well written, but it is more than twice the size of the Results section. The results obtained by the authors are very interesting, and it makes sense to describe them in more detail.

In agreement with reviewer 1, we completed and added some information in the results section. This is an observational study and we feel that discussion is the most important part to raise new questions and open perspectives that lead to new prospective clinical study with detailed virological data. Understanding how genomic evolution is linked to modification of viral fitness as well as infectivity and severity is still one of the main goal to better comprehend viral evolution.  

Point 6: According to the Instructions for authors of VIRUSES, the Conclusions section is not mandatory, but it would be great to add it to the manuscript for clarity.

In agreement with reviewer 1, a conclusion section was added to the manuscript: line 200 to 204.

Point 7: Acknowledgments. This section is unfinished. Please, fix.

In agreement with reviewer 1, the acknowledgement section was completed in the manuscript: line 216 to 218. 

Reviewer 2 Report

Sentis et al show that people infected with the omicron variant of SARS-CoV-2 have lower viral load in nasopharyngeal samples compared to people infected with the delta variant. Thus, the authors conclude that higher transmissibility of omicron is not due to increased viral load in the upper respiratory tract. This is a short but timely study of broad interest. The results are clearly presented and the authors adequately address the limitations of their study.

Minor Points:

  • Please provide information on ethical approval and informed consent.

Author Response

Comments and Suggestions for Authors

Sentis et al show that people infected with the omicron variant of SARS-CoV-2 have lower viral load in nasopharyngeal samples compared to people infected with the delta variant. Thus, the authors conclude that higher transmissibility of omicron is not due to increased viral load in the upper respiratory tract. This is a short but timely study of broad interest. The results are clearly presented and the authors adequately address the limitations of their study.

We thanks the reviewers for these kind comments.

We complete and detailed the methods as the reviewers thought it could be improved and we correct some English thanks to the help of Gregory Queromes.

Minor Points:

  • Please provide information on ethical approval and informed consent.

These informations are related in the Institutional review board statement and Informed Consent Statement line 219 to 227.